# High-Temperature Oxidation Behavior of FeCoCrNi+(Cu/Al)-Based High-Entropy Alloys in Humid Air

**Emma Marie Hamilton White** [1,*], **Mary-Lee Bürckner** [1], **Clara Schlereth** [1], **Maciej Bik** [2] **and Mathias Christian Galetz** [1]

[1] DECHEMA Research Institute, Theodor-Heuss-Allee 25, 60486 Frankfurt am Main, Germany; clara.schlereth@dechema.de (C.S.); mathias.galetz@dechema.de (M.C.G.)

[2] Faculty of Materials Science and Ceramics, AGH University of Krakow, Al. Mickiewicza 30 B8 3.12, 30-059 Krakow, Poland; mbik@agh.edu.pl

[*] Correspondence: emma.white@dechema.de

**Abstract:** Previous studies showed some transition metal high-entropy alloy (HEA) compositions can have good oxidation resistance in air up to 800 °C. Four equiatomic HEAs have been developed based on FeCoCrNi with additions of Mn, Cu, Al or Al+Cu. The oxidation behavior of these HEAs was compared in humid (10 vol.% $H_2O$) air at 800 °C for 100–500 h to investigate the influence of water vapor on the oxidation mechanisms. The Cu- and Al-containing alloys exhibited improved oxidation resistance over the Mn composition. For the Cu-containing alloy, a local attack of the Cu-rich phase was observed, which formed an Fe/Ni/Co/Cr spinel that was surrounded by $Cr_2O_3$. This oxide was thicker for the humid air atmosphere when compared to dry air, and the transition of the Cu oxide to the spinel was accelerated. The Al-containing HEA formed a thin $Al_2O_3$ scale with humidity suppressing AlN formation and forming a smoother oxide layer. The Al+Cu composition had the highest overall oxidation resistance (minimal local attack, no nitridation) and also showed a smooth oxide scale topography under humid air oxidation as opposed to a plate-like, rougher scale under dry air.

**Keywords:** high-entropy alloy; water vapor; oxidation; $Al_2O_3$ formation

## 1. Introduction

High-entropy alloys (HEAs) have demonstrated unique properties, as shown in the innovative work by Cantor et al. [1,2], Yeh et al. [3] and many others [4–6]. One of the most well-known HEAs is the "Cantor" composition, which is an equiatomic FeCoCrNiMn alloy, well characterized for its mechanical, corrosive and oxidation performance [1–4,7–10]. The oxidation behavior of the Cantor composition shows that Mn determines the oxidation rate, rather than Cr [7,9,11]. With Mn, several unprotective oxides form, which have a high density of vacancies, which lead to a fast growth rate [7,9,11]. For other HEA compositions, the available data are still limited in terms of their oxidation and corrosion resistance [12]. Veselkov et al. [13] and Gorr et al. [14] agreed in their recent reviews that Al-containing HEAs are the second most investigated alloys, after Cantor, since Al generally improves high-temperature oxidation performance through alumina scale formation. Oxidation studies of the $Al_x CoCrFeNi$ system by Butler et al. at 1050 °C for 100 h exhibited parabolic oxidation kinetics [15]. If the Al content was high enough, a double-layered oxide structure ($Cr_2O_3$ and $Al_2O_3$) was observed [15], as also confirmed by Hong et al. for $Al_{0.3}CoCrCuFeNi$ [16].

The majority of HEA oxidation studies have been conducted under dry atmosphere conditions; however, humid atmospheres generally have a strong influence on the oxide scale formation of various alloys [17,18]. Kofstadt already described in 1988 [19] that most technical steels oxidize faster in gases containing water vapor than in dry conditions, and

Young [17] provided a comprehensive overview of the different underlying mechanisms occurring in humid atmospheres. These include volatile metal hydroxide formation, gas-scale reactions and oxide density and defect concentration changes (and thus scale growth and scale morphology), as described in detail in [17,19–21]. Additionally, hydrogen adsorption during oxidation into the metal can impact the behavior [22,23].

The behavior of a set of transition metal HEAs—FeCoCrNiCu (HEA+Cu), FeCoCrNiAl (HEA+Al) and FeCoCrNiCuAl (HEA+Cu/Al)—was previously investigated under metal dusting conditions [24]. At 620 °C in a low-oxygen partial pressure atmosphere, the Al-containing alloys had thermally grown $Al_2O_3$ or $Cr_2O_3$ scales and effectively inhibited carbon ingress [24]. These alloys, plus the Cantor composition for comparison, were also studied at higher temperatures of 600–800 °C for up to 500 h in synthetic air to elucidate their oxidation mechanisms [25]. Of even further scientific and industrial interest is the performance of this set of transition metal HEAs under humid atmospheres. Therefore, in this paper, the HEA+Cu, HEA+Al and HEA+Cu/Al oxide compositions, morphologies and subsurface phenomena at 800 °C in humid air for up to 500 h are investigated and compared to dry air. Additionally, the Cantor composition was included for comparison, since generally the influence of water vapor on the oxide scale formation is not well understood for these novel compositions.

## 2. Experimental Methods

Arc-melted button samples were produced by the U.S. Department of Energy's Ames National Laboratory (Ames, IA, USA) using an electric arc furnace on a water-cooled Cu hearth. To ensure alloy homogeneity, each button was flipped and remelted three times. The buttons were cut using wire electro-discharge machining into samples sized $3 \times 4 \times 7$ mm$^3$. These were then ground to P800 grit with SiC paper and cleaned with acetone in an ultrasonic bath.

The samples were placed in a quartz glass tube in a six-zone furnace where the tube was flooded with dry or humid synthetic air (80N$_2$-20%O$_2$, +10 vol.% H$_2$O, 4.8 L/h) at 800 °C for 100, 300 and 500 h. After being removed from the furnace, the exposed samples were cleaned, Au-sputter-coated, placed in a Ni electrochemical bath and then embedded for cross-sectional characterization. Most samples were hot-embedded; however, those samples that showed a more porous oxide scale, likely prone to spallation, were chemically embedded to preserve the adherence of the oxide scale for characterization.

The sample microstructures were imaged with light microscopy and scanning electron microscopy (SEM, HITACHI SU1000, Tokyo, Japan), equipped with energy dispersive spectroscopy (EDS). The oxide scales and subsurface microstructures were characterized qualitatively and quantitatively using an electron probe microanalysis (EPMA, JXA-8100 manufactured by Jeol (Akishima, Japan)). X-ray diffraction (XRD) measurements were performed in order to identify the thermally grown oxide scales of all HEAs. Unfortunately, the results were unsatisfactory due to the high fluorescence of the bulk materials. Therefore, Raman spectroscopy was pursued instead to identify the oxide scale compositions. A WITec alpha 300 M+ spectrometer with a 488 nm laser line and laser spot of ca. 650 nm in diameter, a 100× ZEISS Epiplan-Neufluar (ZEISS, Jena, Germany) objective with NA = 0.9, 600 grooves/mm grating and an Andor CCD detector (Oxford Instruments, Abingdon, UK) were used to determine phase compositions. The scanning area during mapping was varied depending on the dimensions of the oxide scales. Single accumulations lasted either 1 s ($Al_2O_3$ detection) or 2 s (detection of all other phases) and the sampling density was equal to 500 nm in each case. WITec Control FIVE was used to collect the data, and subsequently both the WITec Project FIVE 5.3 PLUS and OPUS 7.2 software (Oxford Instruments, Abingdon, UK) were used to post-process the recorded spectra. First, the selected ranges (110–1200 cm$^{-1}$ and 5000–7200 cm$^{-1}$) were extracted and cosmic spikes were removed using the CRR filter. Then, integration was carried out using a special filter of predefined width and position depending on the targeted analyzed phase, e.g., the 555 cm$^{-1}$ band for $Cr_2O_3$. Component concentration maps were generated, as well as

the most representative spectra for each phase that were extracted manually. Distribution images were assembled to create a combined distribution image that was overlaid with the confocal image. For all Raman results, the distribution images of the phases (based on integration of the characteristic band with the Raman shift given in the brackets) are presented with the corresponding spectra.

## 3. Results

The as-solidified chemical compositions and microstructures were previously determined and discussed and are listed in Table 1 [24,25].

**Table 1.** Chemical compositions in wt.% (from EDS) and microstructural descriptions of the investigated Al- and Cu-containing HEAs (adapted from [24,25]).

| Alloy | Cu | Co | Cr | Fe | Ni | Al | Mn | Microstructure |
|---|---|---|---|---|---|---|---|---|
| HEA+Cu | 21.9 | 20.4 | 17.9 | 19.3 | 20.3 | - | - | FCC+FCC Cu-rich (45 at.%) |
| HEA+Al | - | 23.3 | 20.6 | 22.1 | 23.2 | 10.7 | - | interdendritic BCC (18.8 at.%Al, 23.7 at.%Cr) and FCC (12.8 at.%Al, 29.2 at.%Cr) |
| HEA+Cu/Al | 20.1 | 18.6 | 16.4 | 17.7 | 18.6 | 8.5 | - | BCC+FCC Cu-rich (up to 66 at.%) |

### 3.1. Cantor (HEA+Mn)

The single phase FCC Cantor composition has been extensively discussed in terms of dry air oxidation resistance elsewhere [6,7,9,11,12,25] and so is only briefly included as a baseline in the current study. HEA+Mn forms a thick oxide at 800 °C for up to 500 h as seen in Figure 1. The oxide scale is porous and irregular, spalls locally and contains three layers: an exterior Mn-rich oxide, a Cr/Co/Fe/Co-rich spinel-type oxide and a thin $Cr_2O_3$ layer at the oxide/substrate interface as evidenced by the EPMA results in Figure 2. Mn (and to a lesser extent Cr) is depleted in the metal subsurface zone due to fast outward diffusion leaving Kirkendall voids. Under humid air, the initial oxide layer appears to be thinner, but simultaneously the presence of water vapor increases outward diffusion as evidenced by the increased Kirkendall porosity in the subsurface microstructure. Additionally, some internal Mn oxidation appears, as evidenced by the internal Mn depletion in the EPMA of Figure 2. Chromia also appears to form around the internal Mn oxides when connected with the external scale (Figure 2).

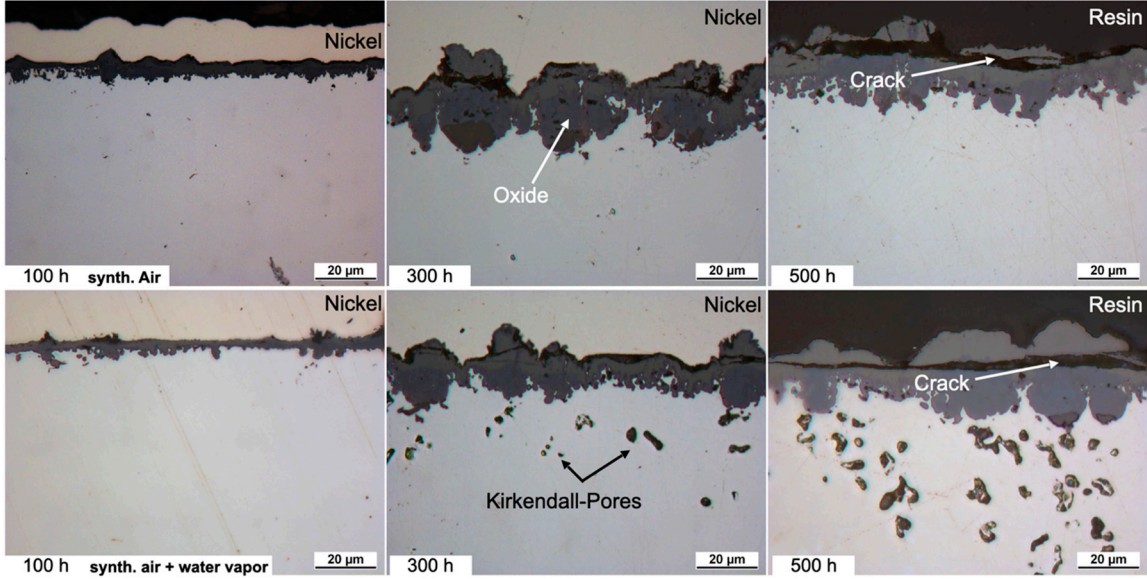

**Figure 1.** Light microscope images of the cross-sections of HEA+Mn after exposure at 800 °C for 100, 300 and 500 h in dry air and air + 10 vol.% $H_2O$. Dry-air images from [25].

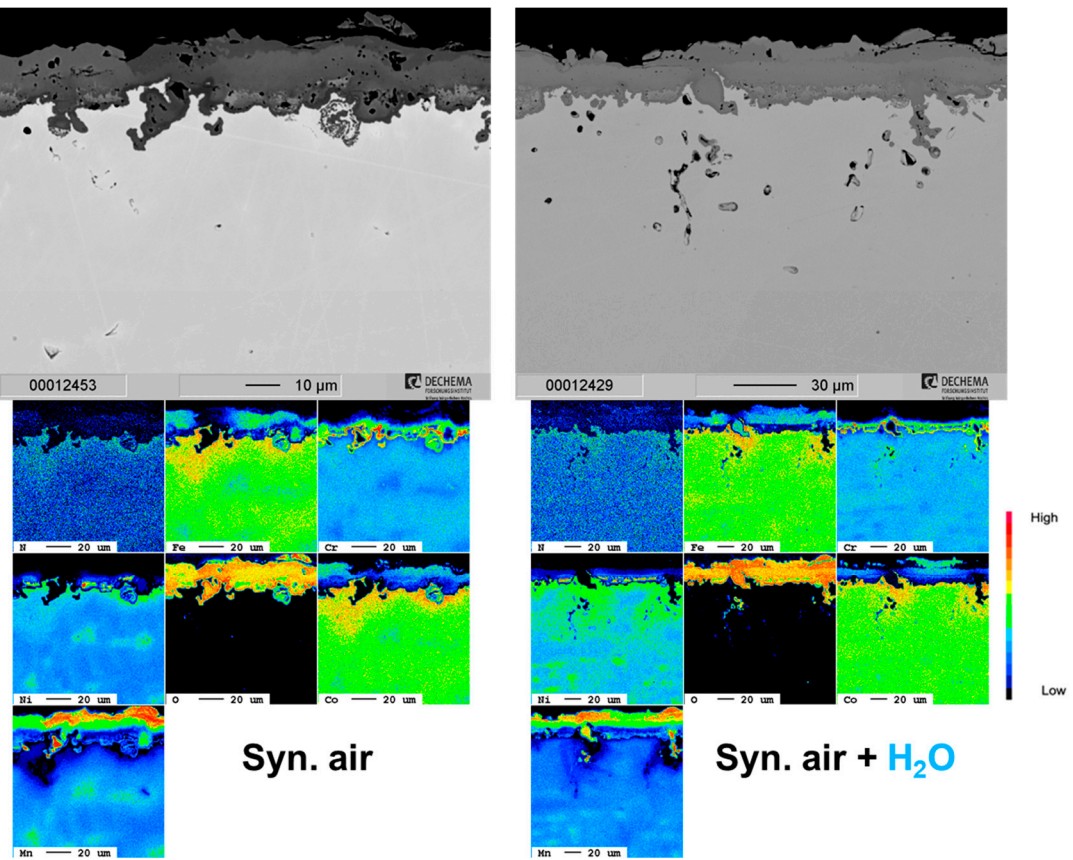

**Figure 2.** Cross-section SEM images and corresponding EPMA element maps of the HEA+Mn in dry and humid air at 800 °C after 500 h.

### 3.2. HEA+Cu

A selective attack of the Cu-rich phase of the HEA+Cu is prominent at all exposure times and under both dry and humid air as shown in Figure 3. Additionally, blisters formed on top of the surface preferentially over the Cu-rich phase with apparent Cu outward diffusion and then subsequent oxidation. The remaining surface formed a thin Cr-rich oxide scale, which after 500 h becomes thick enough that the oxidized blisters then become indistinguishable. The depth of the internal oxidation of the Cu-rich phase increases with water vapor exposure as is clearly shown in the elemental maps of Figure 4. The former Cu-rich phase converts to a Cr/Co/Fe/Ni-rich spinel-type oxide with $Cr_2O_3$ at the prior interface, mitigating further oxygen penetration [25].

Figure 5 (dry) and Figure 6 (humid) include Raman spectra and corresponding component distribution maps identifying the formed spinels and other oxides for the HEA+Cu exposed in dry and humid air after both 100 h and 300 h. The labeled Fe doping in $CuCr_2O_4$ is based on the high intensity of the ca. 540 ÷ 545 cm$^{-1}$ band versus the band at ca. 685 ÷ 695 cm$^{-1}$ [26]. The labelled Co doping of the NiO is based on the combined EPMA results and the shift of the main Raman band from ca. 510 to 530 cm$^{-1}$—although, e.g., Cu and Fe could also be involved since any cation of a lower radius than Ni shifts the main band toward higher values [27–29]. The NiFe$_2$O$_4$ had no or very minimal Co doping as the bands at high positions are characteristic for NiFe$_2$O$_4$ with the main one at ca. 706 cm$^{-1}$ and the lower intensity band at 489 cm$^{-1}$. The labeled Fe doping of the CoCr$_2$O$_4$ and CuCr$_2$O$_4$ for the HEA+Cu is based on cross-correlating with the EPMA maps. Based on the literature, Fe should substitute Cr in small amounts: $x \leq 0.1$ for the CoCr$_{2-x}$Fe$_x$O$_4$ composition [30,31] as well as the CuCr$_{2-x}$Fe$_x$O$_4$ composition, which is in line with the highest intensity of the A$_{1g}$ band at ca. 690 cm$^{-1}$ [26].

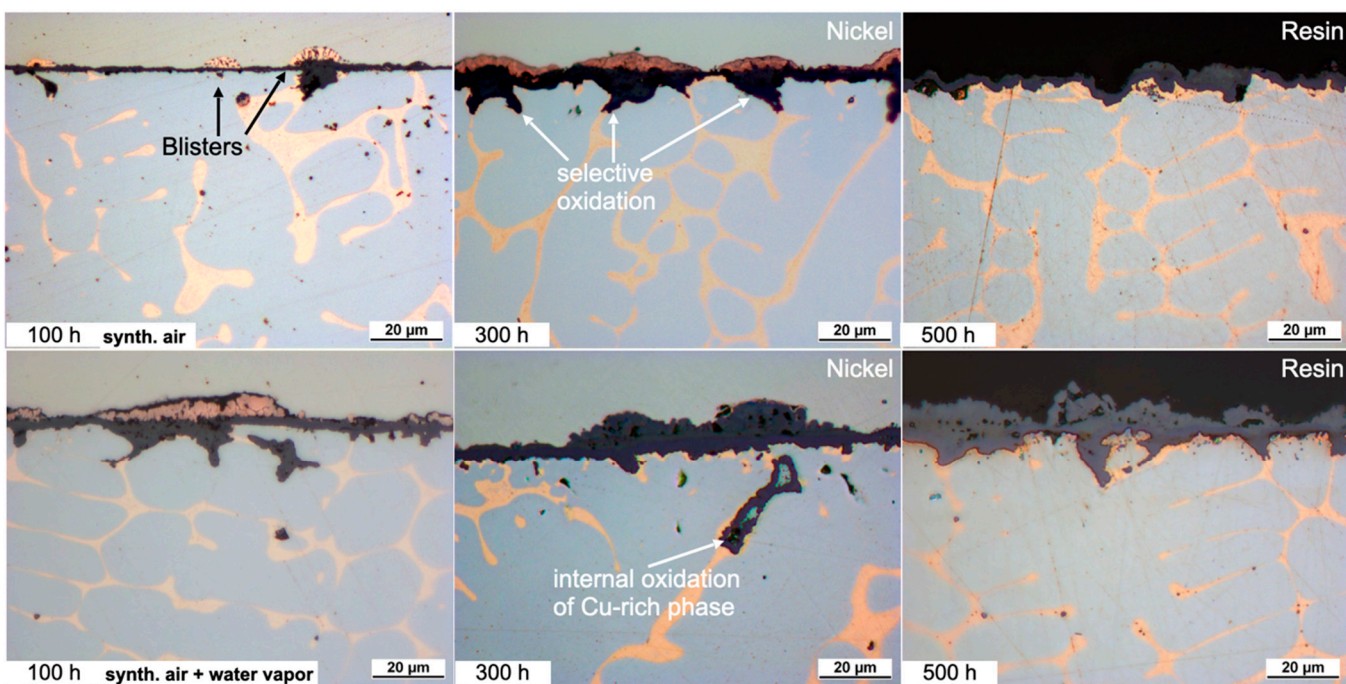

**Figure 3.** Light microscope images of the cross-sections of the HEA+Cu after exposure at 800 °C for 100, 300 and 500 h. Dry-air images from [25].

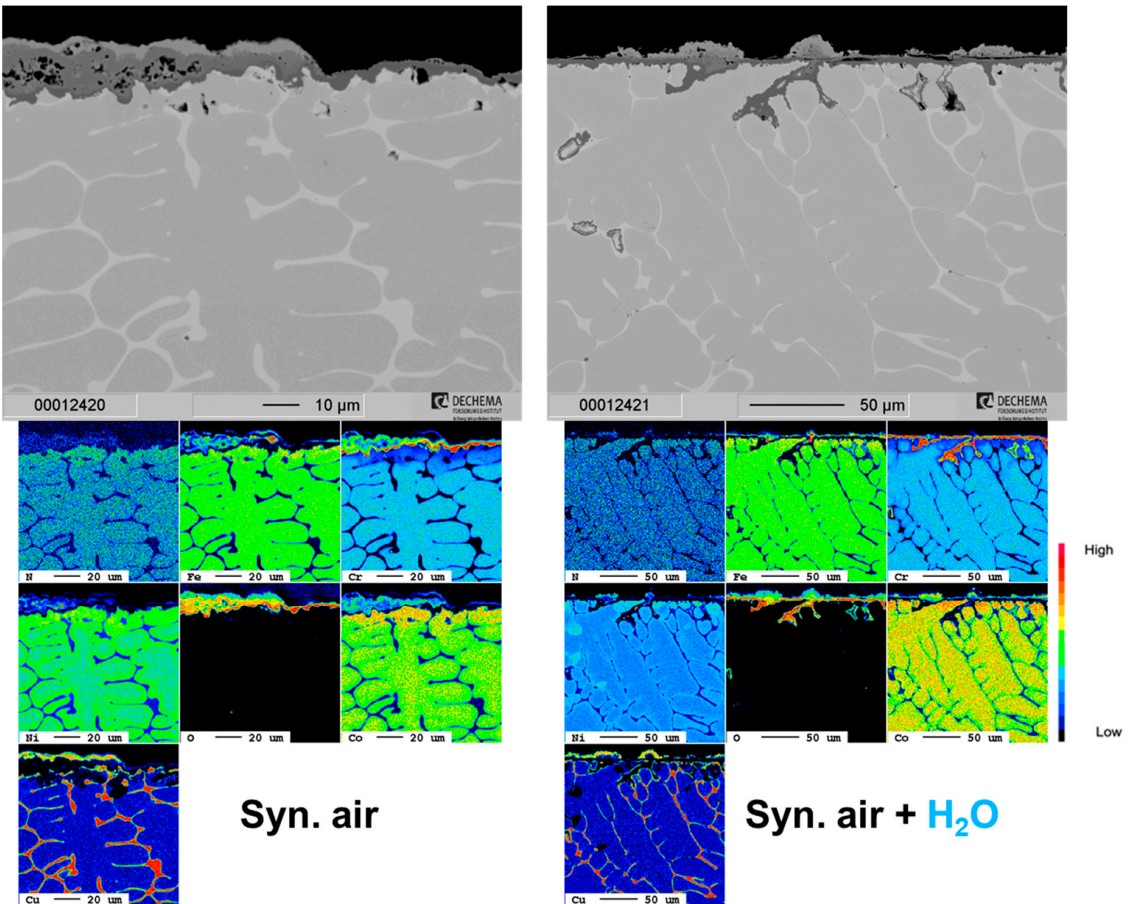

**Figure 4.** Cross-section SEM images and corresponding EPMA element maps of the HEA+Cu in dry and humid air at 800 °C after 500 h.

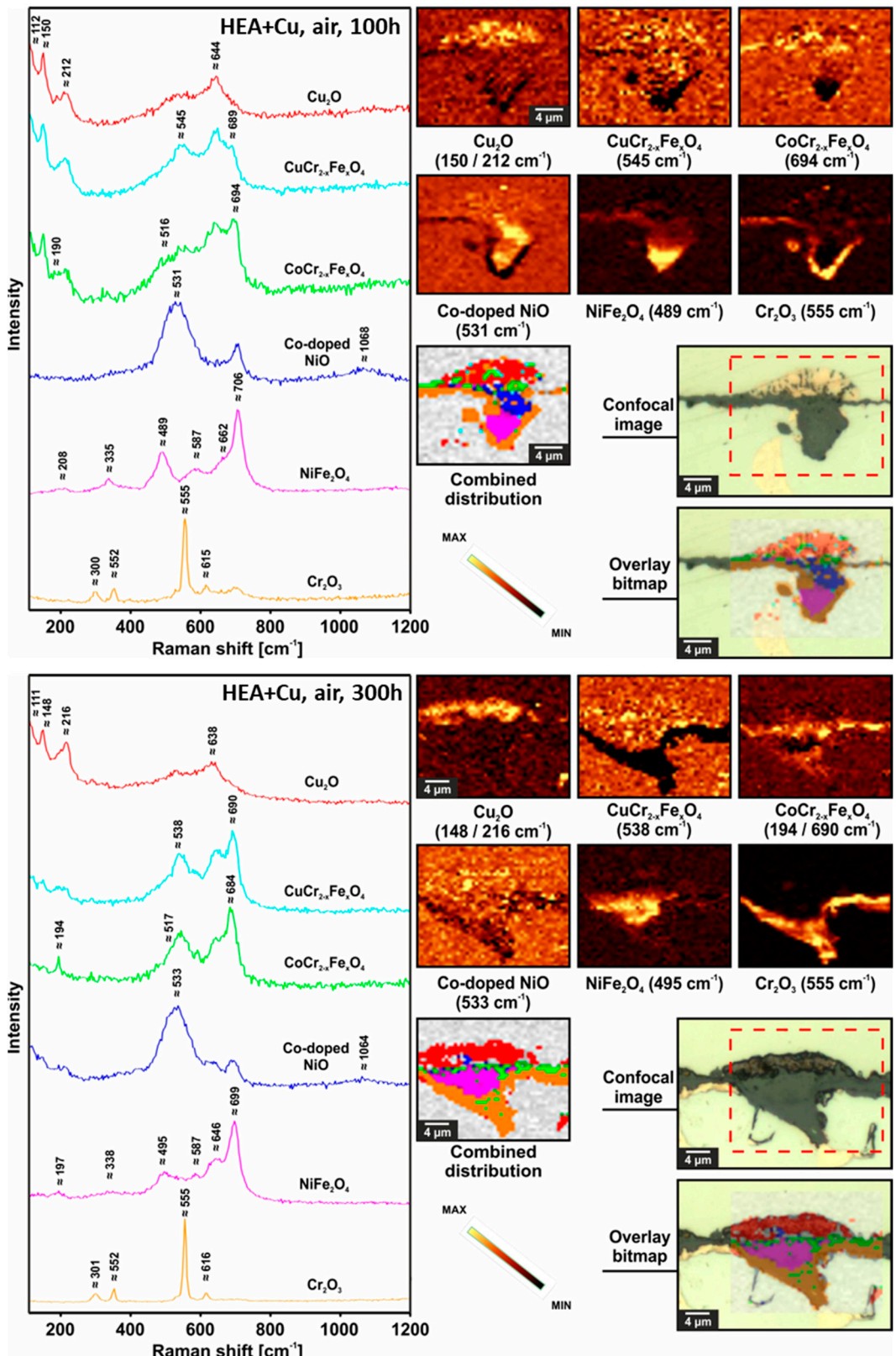

**Figure 5.** Raman confocal imaging for HEA+Cu after exposure at 800 °C for 100 h in dry air (**top**) and for 300 h in dry air (**bottom**). For the 300 h results, the investigated area was chosen from the optical image presented in Figure 3, and the Raman distribution image for CuCr$_{2-x}$Fe$_x$O$_4$ spinel was not taken into account when preparing combined distribution image due to its very small concentration.

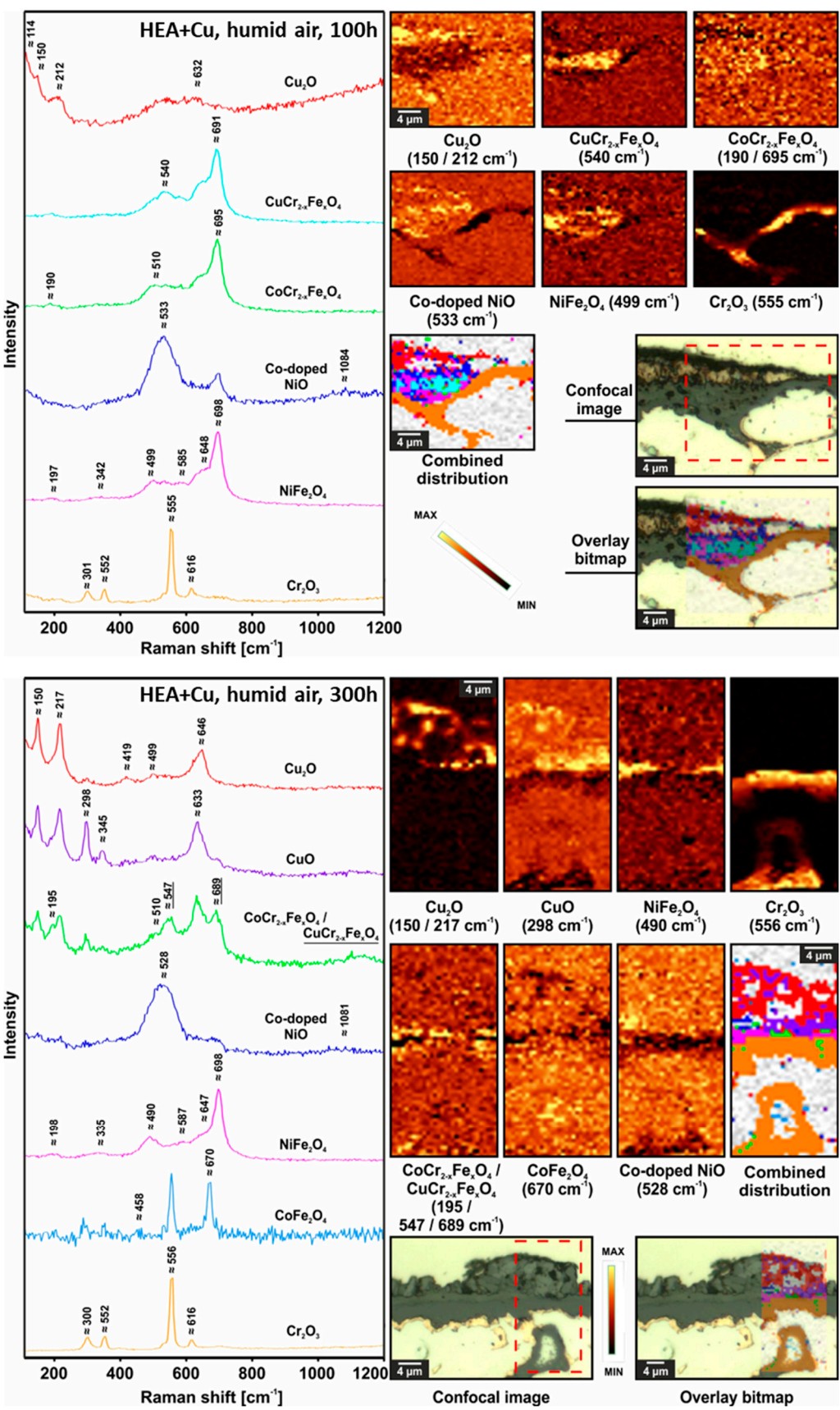

**Figure 6.** Raman confocal imaging for HEA+Cu after exposure at 800 °C for 100 h in humid air (**top**) and for 300 h in humid air (**bottom**). The investigated areas were chosen from optical images presented in Figure 3.

The Raman results highlight differences between 100 h and 300 h for the HEA+Cu exposed to dry air at 800 °C as oxidation progresses. After 300 h, there is more $Cr_2O_3$, less NiO and more Fe in the CoCr spinel (shift of the $A_{1g}$ band toward lower Raman shift values [31]), less CuCrFe spinel and most likely a higher concentration of $Cu_2O$. Compared to the dry-air samples at the same time points, under humid air, the HEA+Cu samples after 100 h and 300 h have less $Cu_2O$ as Cu had further oxidized to CuO. The internal oxidation evident after 300 h of oxidation in humid air contains $Cr_2O_3$ as the main constituent and a minor quantity of the spinel, most likely $CoFe_2O_4$.

### 3.3. HEA+Al

For the HEA+Al, a thin, homogeneous, continuous $Al_2O_3$ layer forms after 100 h in both dry and humid air (Figure 7). With continued exposure after 300 h in dry air, an Al depletion zone is evident with some N uptake and internal $Al_2O_3$ oxidation, so the entire scale becomes thicker, more needle-like and less uniform. Qualitative and quantitative measurements with EPMA (Figure 8) indicate the internal attack is related to the formation of AlN [25]. However, this scale appears stable as no further significant changes can be detected up to 500 h in dry air. In humid air, on the other hand, the nitride formation is less pronounced and the alumina layer remains protective and thin at 300 h and is only slightly thicker after 500 h with a much thinner Al depletion zone beneath the scale when compared to dry air.

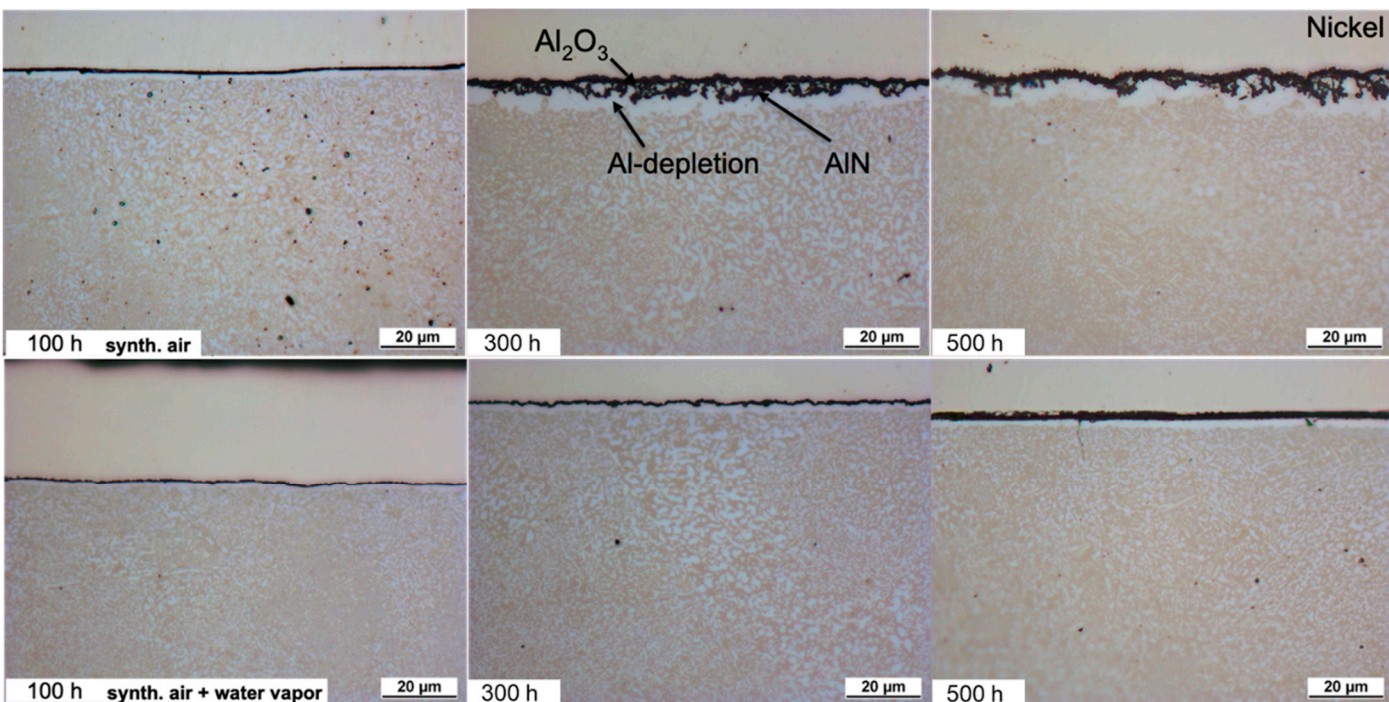

**Figure 7.** Light microscopic images of the cross-sections of HEA+Al after exposure at 800 °C. Dry-air images from [25].

According to Figure 9, the Raman spectra and corresponding component distribution maps showed that after 500 h under dry air the HEA+Al formed more $Cr_2O_3$ and $\alpha$-$Al_2O_3$ (not fully crystallized/transformed from metastable alumina based on the Raman band contour), as well as $\theta$-$Al_2O_3$. Such phases are not uniformly distributed across the surface. Under humid air after 500 h, no $\theta$-$Al_2O_3$ could be observed in the continuous external scale, and the $\alpha$-$Al_2O_3$ was less crystallized based on its broad Raman band contour. For humid air, a greater share of Fe in the CoCr spinel could be observed based on the shift of the $A_{1g}$ band toward lower Raman shift values [31]. The EPMA results on nitridation were confirmed. Based on different intensity ratios of the characteristic bands coming

from AlN, one can distinguish between the different crystallographic planes exposed to the Raman laser—either 100 or 002—which was also described in [32], showing that there is no preferred orientation of the internally grown AlN.

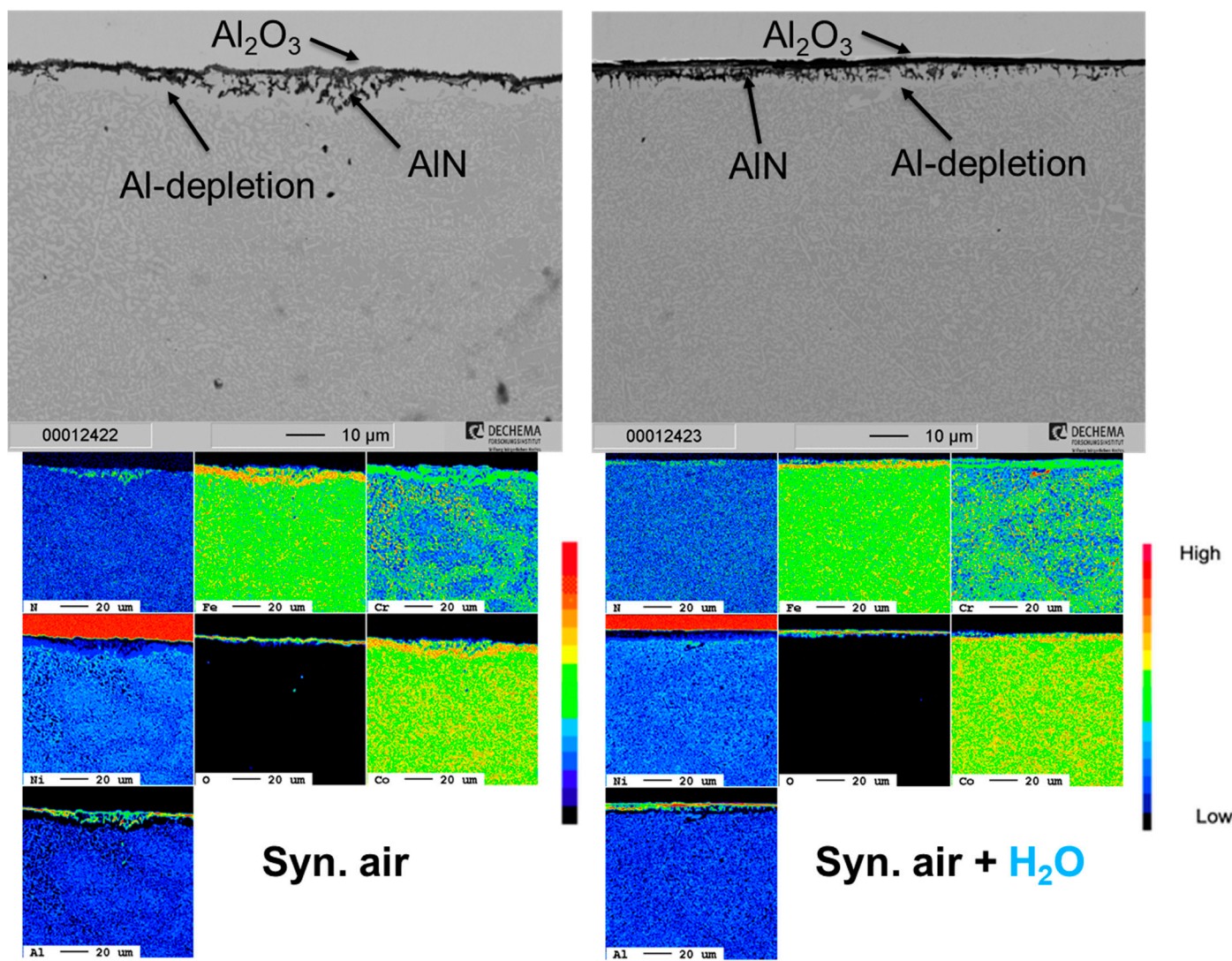

**Figure 8.** Cross-section SEM images and corresponding EPMA element maps of the HEA+Al in dry and humid air at 800 °C after 500 h. Synthetic air adapted from [25].

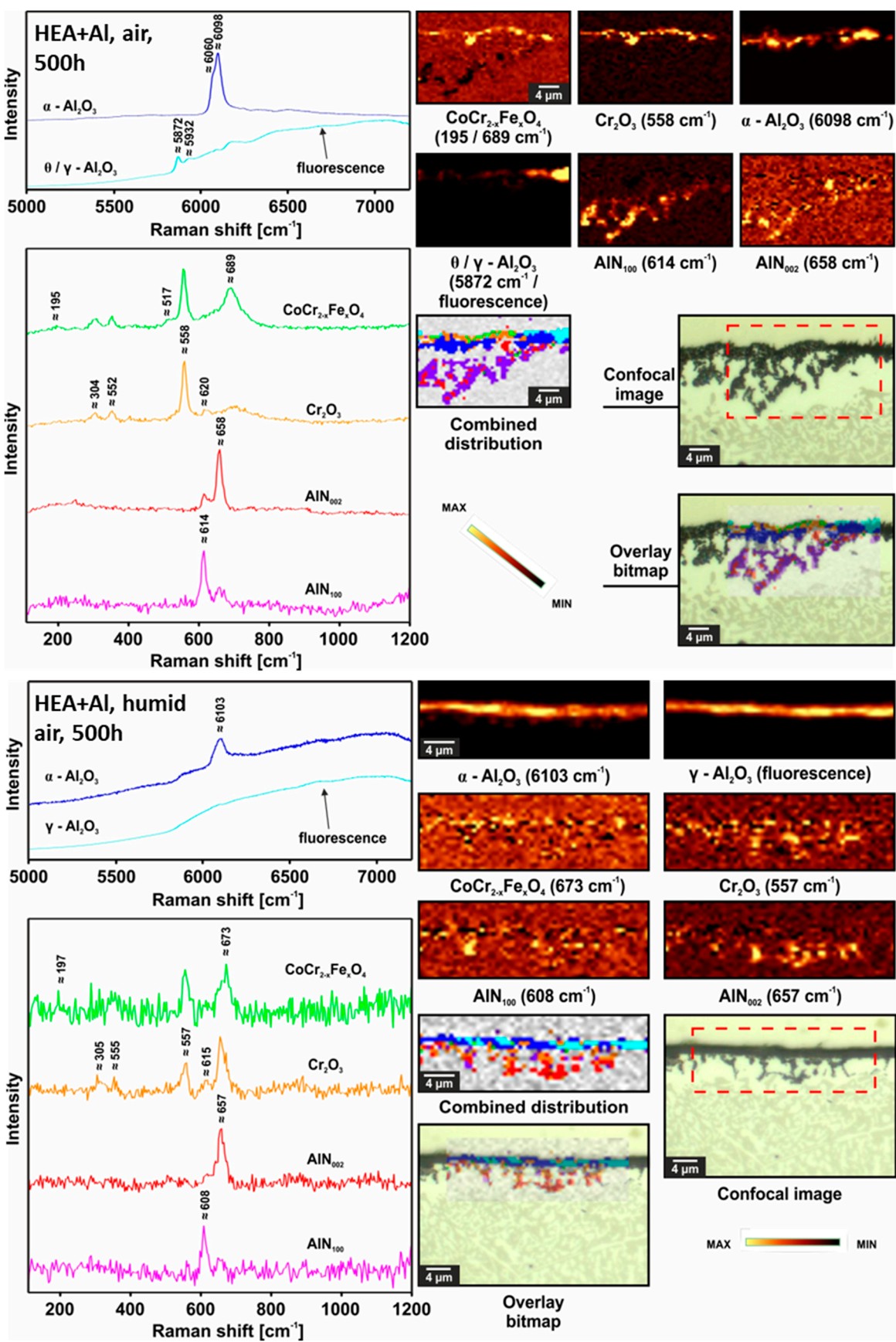

**Figure 9.** Raman confocal imaging for HEA+Al after exposure at 800 °C for 500 h in dry air (**top**) and in humid air (**bottom**). The investigated areas were chosen from the SEM images presented in Figure 6.

### 3.4. HEA+Cu/Al

Similar to the HEA+Al, HEA+Cu/Al forms a continuous, thin $Al_2O_3$ layer after 100 h at 800 °C in both dry and humid synthetic air as seen in Figure 10. A local attack of the Cu-rich phase (as seen in HEA+Cu) was greatly inhibited, occurring only minimally where larger regions of the Cu-rich phase were directly at the surface. Some Al depletion was observable beneath the oxide scale of the HEA+Cu/Al in both atmospheres, which increases with further exposure time. In contrast to the HEA+Al, no formation of AlN nor extensive internal oxidation could be observed as confirmed in the EPMA maps of Figure 11 and the Raman measurements of Figure 12. In the case of dry air exposure, the HEA+Cu/Al formed metastable alumina phases (θ and γ) and $(Co,Ni)Fe_2O_4$ spinel in the small regions of a local attack after 500 h at 800 °C. For humid air exposure, the HEA+Cu/Al forms an external $γ-Al_2O_3$ scale with some θ-alumina, but less than for dry air. Additionally, no spinels were evident for the humid air exposure. Some minor $Cr_2O_3$ and co-located $α-Al_2O_3$ formation occurred internally after the humid air atmosphere. This occurred most likely in pores within the sample. No Cu oxides were present for either atmosphere, nor were any signs of nitridation found for the HEA+Cu/Al. The oxide scale remains especially thin and smooth for the humid air environment, while the dry synthetic air exposure shows a needle-like scale morphology after 300 h and 500 h. This topographical difference is further highlighted in the higher-magnification SEM images of the 300 h exposures in both atmospheres in Figure 13.

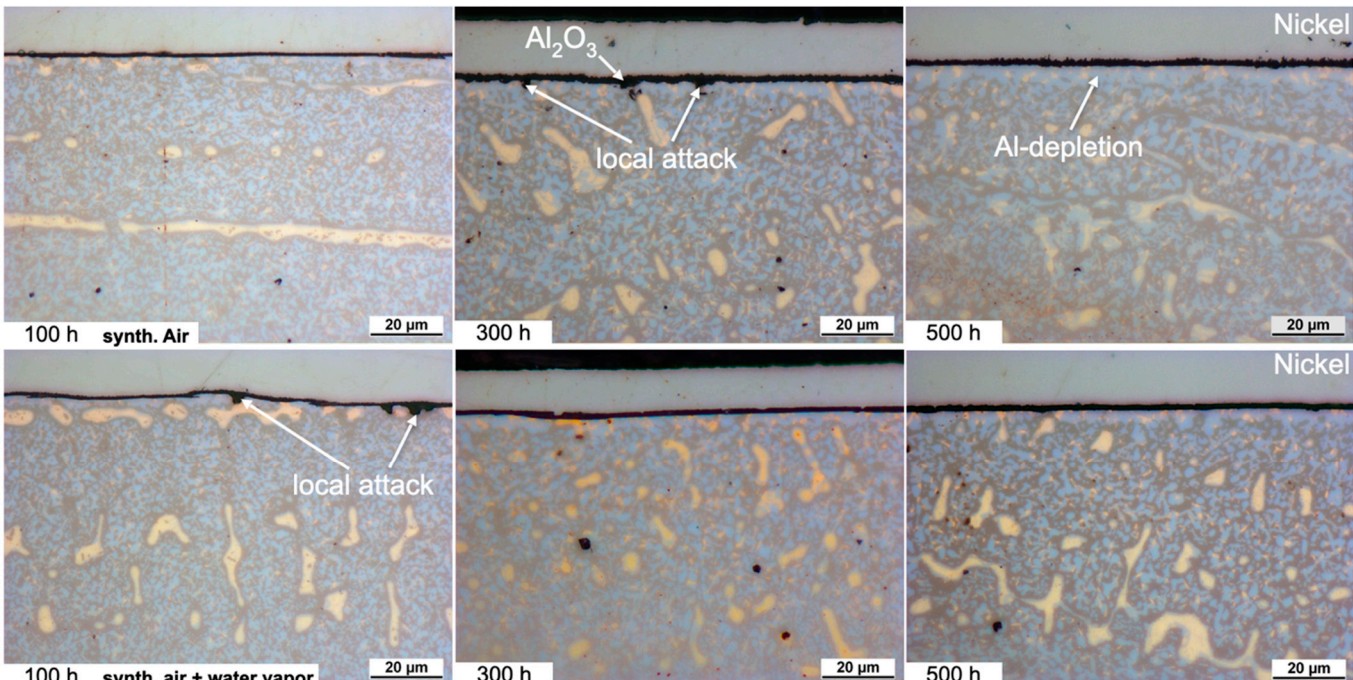

**Figure 10.** Light microscopic images of the cross-sections of HEA+Cu/Al after exposure at 800 °C. Dry-air images from [25].

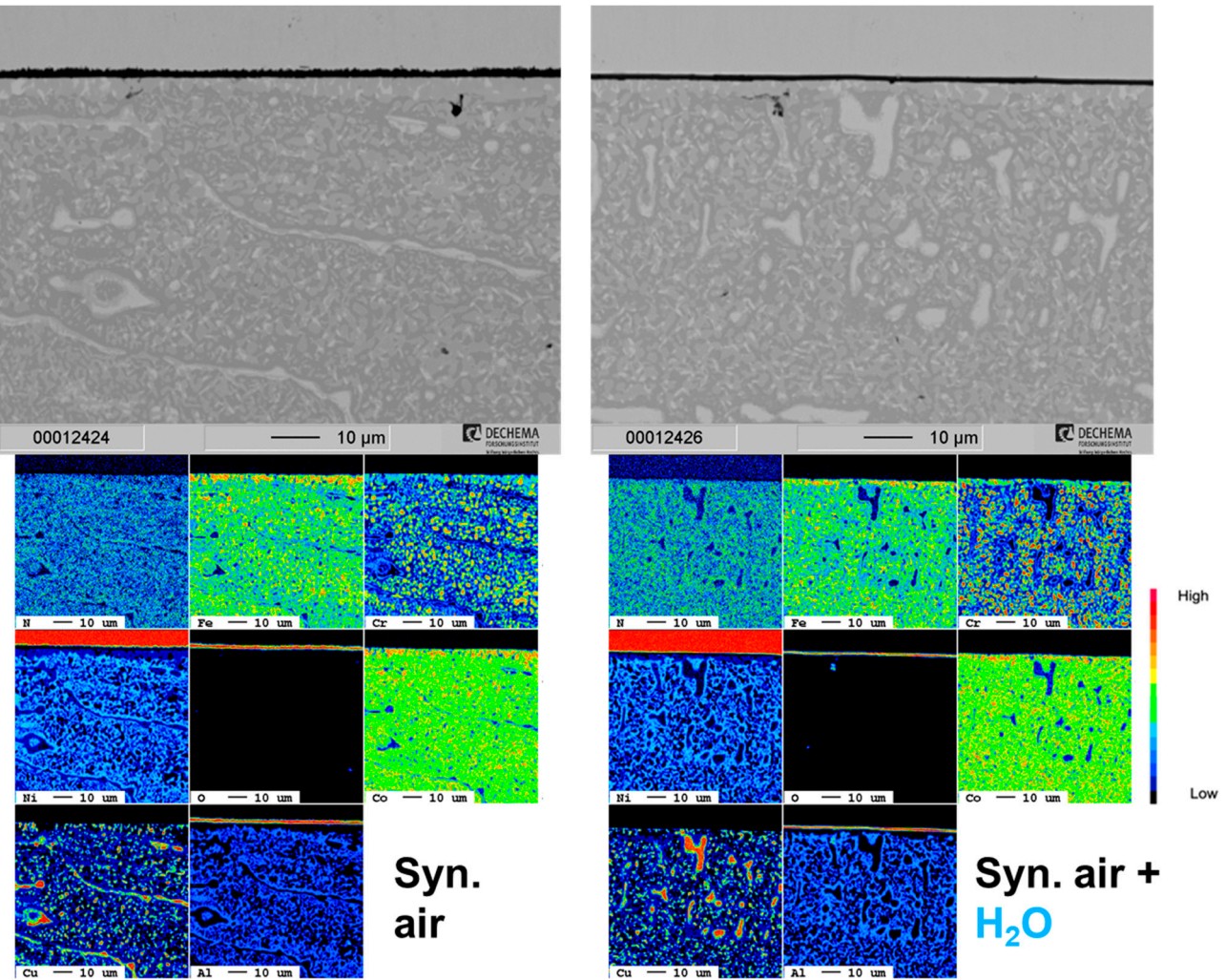

**Figure 11.** Cross-section SEM images and corresponding EPMA element maps of the HEA+Cu/Al in dry and humid air at 800 °C after 500 h. Synthetic air adapted from [25].

For the humid-air results, the Raman distribution image for $Cr_2O_3$ was not taken into account when preparing the combined distribution image due to its very small concentration.

While some surface porosity prevented mass gain measurements for these samples, the average oxide scale thicknesses were measured (using SEM images) for an approximation of the oxidation kinetics. Figure 14 shows the scale thickness versus time for the air and humid air exposures for the three alloys HEA+Cu, HEA+Al and HEA+Cu/Al. The HEA+Cu clearly has the thickest scale in both atmospheres, while the Al-containing HEAs have much thinner scales with the thinnest forming under water vapor exposure.

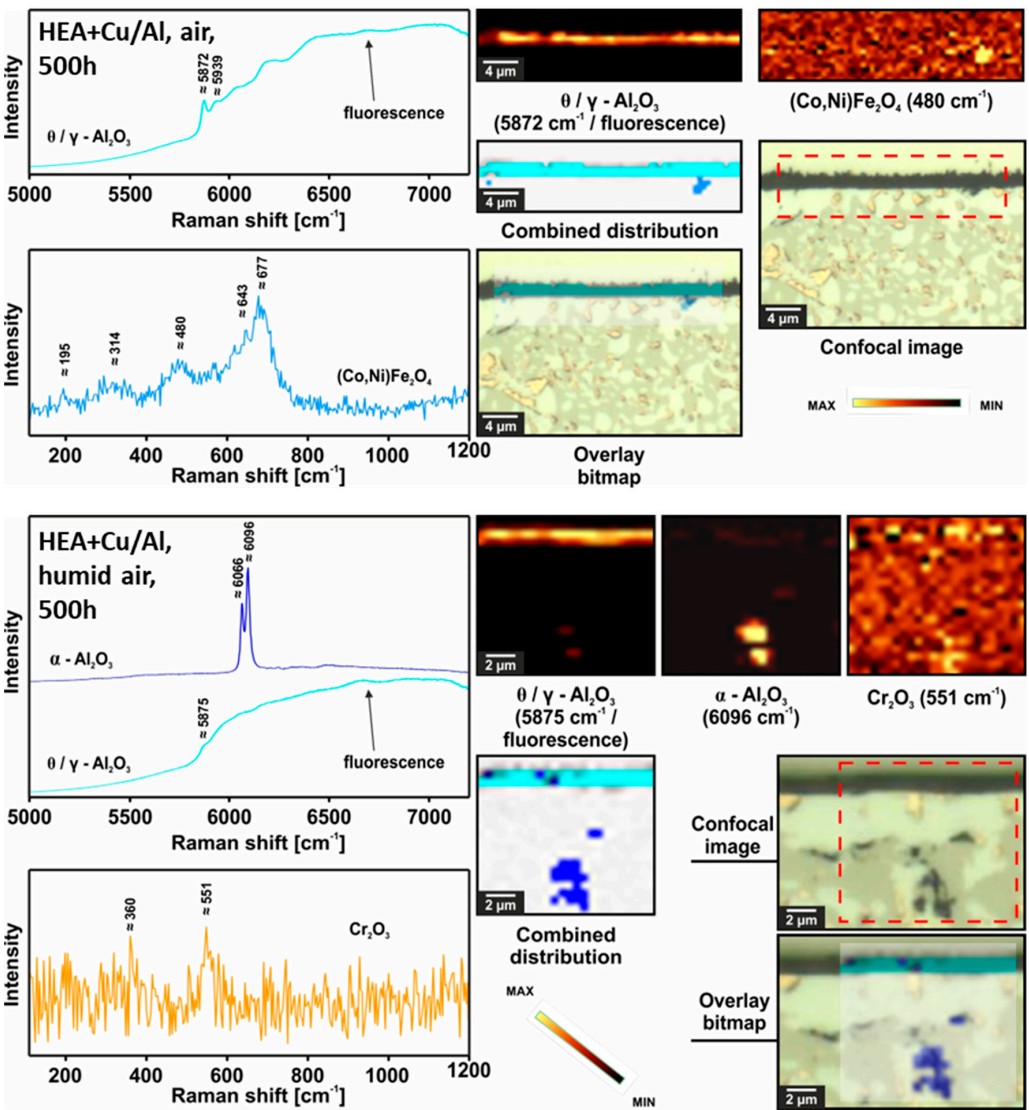

**Figure 12.** Raman confocal imaging for HEA+Cu/Al after exposure at 800 °C for 500 h in dry air (**top**) and humid air (**bottom**). The investigated areas were chosen from the SEM images presented in Figure 8.

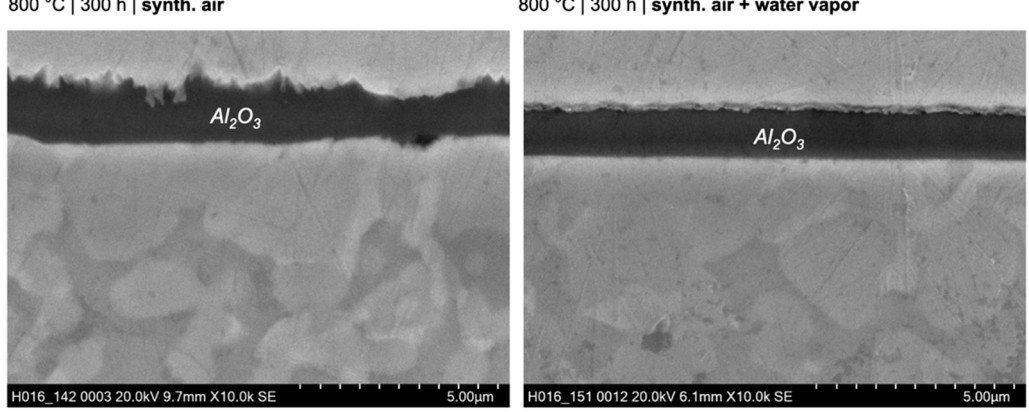

**Figure 13.** Scanning electron microscopy images of the morphology of HEA+Cu/Al after exposure for 300 h at 800 °C in dry and humid synthetic air.

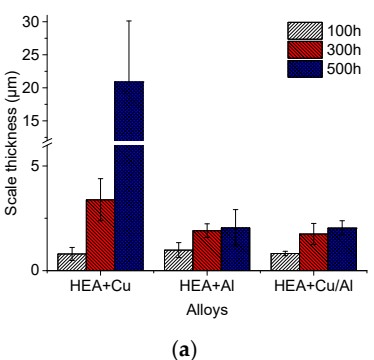
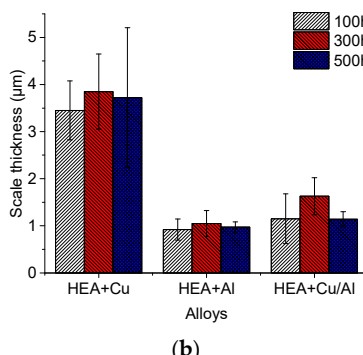

(a)  (b)

**Figure 14.** Oxide scale thicknesses of HEA+Cu, HEA+Al and HEA+Cu/Al after exposure for 100, 300 and 500 h at 800 °C in dry (**a**) and humid (**b**) synthetic air.

## 4. Discussion

As already mentioned for water vapor, complicated effects on high-temperature oxidation behavior were found including increasing proton defects in the oxide scale [33,34], accelerating the transformation of θ- to α-$Al_2O_3$ [35], forming $H_2$–$H_2O$ (facilitating rapid inward transport of O across pores) [36,37] and increasing the vaporization of some oxides by forming hydroxides [35]. The relevant hydroxides and the partial pressures of each hydroxide for $p_{H2O}$ = 1 atm and $p_{O2}$ = 0.5 atm at 600 °C and 1200 °C are included in Table 2 [38,39]. Here, it is clear that the Cr hydroxide is by far the most volatile, followed by the hydroxides of Ni, Co, Fe, Mn and finally Al as temperatures increase [38]. Experimentally, it was confirmed that volatile Cr hydroxide formation from chromia is the most severe in these types of air plus water vapor conditions [40]. Beyond volatilization, water vapor has been shown to refine the grains of $Cr_2O_3$ scales, and the growth kinetics are simultaneously accelerated by grain boundary diffusion of OH- or $H_2O$ [41]. These mechanisms and effects are discussed separately for each alloy in the following.

**Table 2.** Selected thermodynamic data for hydroxide vapor species, partial pressures of each hydroxide for $p_{H2O}$ = 1 atm and $p_{O2}$ = 0.5 atm at 600 °C and 1200 °C. Adapted from [38,39].

| Reaction | 600 °C | 1200 °C |
|:---:|:---:|:---:|
| $0.5Cr_2O_3(c) + H_2O(g) + 0.75O_2(g) = CrO_2(OH)_2(g)$ | $1.97 \times 10^{-6}$ | $4.40 \times 10^{-5}$ |
| $0.5Mn_2O_3(s) + H_2O(g) = Mn(OH)_2(g) + 0.25O_2(g)$ | $1.57 \times 10^{-15}$ | - |
| $0.333Mn_3O_4(s) + H_2O(g) = Mn(OH)_2(g) + 0.167O_2(g)$ | - | $1.62 \times 10^{-7}$ |
| $0.5Fe_2O_3(c) + H_2O(g) = Fe(OH)_2(g) + 0.25O_2(g)$ | $7.01 \times 10^{-15}$ | $3.46 \times 10^{-7}$ |
| $0.333Co_3O_4(c) + H_2O(g) = Co(OH)_2(g) + 0.167O_2(g)$ | $1.04 \times 10^{-12}$ | - |
| $CoO(c) + H_2O(g) = Co(OH)_2(g)$ | - | $4.70 \times 10^{-6}$ |
| $NiO(c) + H_2O(g) = Ni(OH)_2(g)$ | $1.56 \times 10^{-11}$ | $5.10 \times 10^{-6}$ |
| $0.5Al_2O_3(c) + 1.5H_2O(g) = Al(OH)_3(g)$ | $3.87 \times 10^{-12}$ | $1.42 \times 10^{-7}$ |

### 4.1. HEA+Mn

The oxidation mechanisms observed here for the HEA+Mn (Cantor) are in good accordance with the literature [6,7,9,11]. Poor oxidation resistance is primarily characterized by the fast Mn-oxide growth rates leading to severe Mn depletion and formation of Kirkendall voids in the metal subsurface [7,9,11,17]. Mn outward diffusion is ~2 orders of magnitude higher than Cr, Ni and Co [19]. Additionally, oxygen inward diffusion through the outer Mn oxide is relatively high due to the high number of vacancies in the oxide [42].

Under water vapor exposure, the Kirkendall voids and internal oxidation of Mn were more prevalent, although the oxide scale remained thinner, at least after 100 h and 300 h at 800 °C. Extensive scale cracking and spallation can be seen in the cross-sections of Figure 1,

for both dry and humid air. Nevertheless, Stephan-Scherb et al. [11] recently proposed the oxide composition and the oxidation state of the Mn oxide could vary under humid air. As an Fe/Co/Cr-spinel forms on the HEA+Mn alloy [25], which does not look too different in humid air, the evaporation of Cr as a Cr hydroxide under a humid air atmosphere is also likely increasing the outward diffusion and subsequent void formation [41]. Some suppression of Cr volatilization has been observed through outer Mn oxide scale formation, which could be why the humid- and dry-air scales are similar after 500 h due to the similar behavior of the manganese chromium spinel in both atmospheres [43]. Eventually, Cr does enrich in the interior of the oxide scale due to the higher outward diffusion of Mn, and even Fe and Co. However, due to the highly local variation in the oxide scale composition, the alloy is only partially protected, especially with further oxide spallation. A schematic of the oxidation process under humid air is shown in Figure 15.

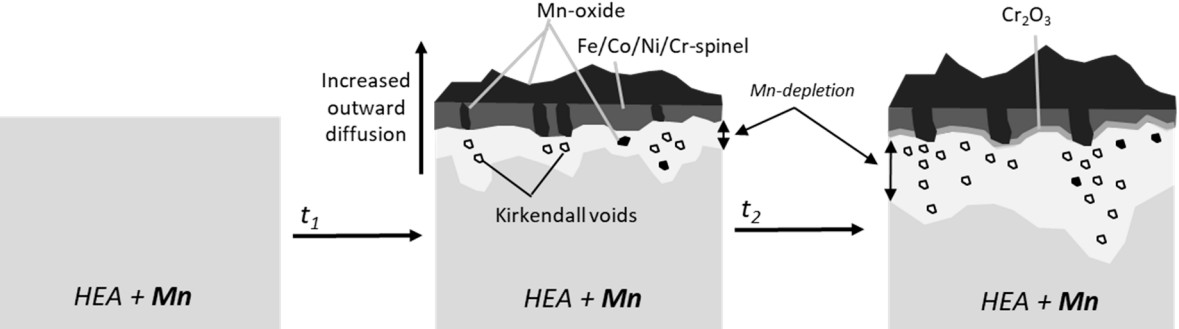

**Figure 15.** Proposed oxidation mechanism of HEA + Mn as a function of time under humid air exposure at 800 °C.

### 4.2. HEA+Cu

The two-phase microstructure of the HEA+Cu greatly affects the oxidation behavior as the Cu-rich and Cu-lean phases have varying oxidation resistance [25,44–46]. While a triplex oxide scale consisting of an external Cu oxide, an Fe/Co/Cr-spinel interlayer and an interior Cr-rich oxide layer at the metal oxide interface eventually forms, the initial, transient stage above the surface-connected Cu-rich phase is complex, as has been previously described [25,45]. Still, the elimination of Mn and addition of Cu to the alloy provide better oxidation protection in the form of a thinner oxide layer in both dry and humid atmospheres at 800 °C than what was observed for the HEA+Mn. Without Mn, Cr can act more effectively as a protective oxide forming element in the HEA system. However, the observed selective attack of the Cu-rich phase serves as a diffusion path for oxygen and hence facilitates further oxidation.

The presence of water vapor appears to accelerate outward diffusion of Cu and Cr, as not only are the blisters larger, but also after only 100 h there is a thin external scale over the Cu-rich blisters, while after 300 h the blisters are already difficult to differentiate from the other oxides (Figure 3). Additionally, the depth of internal oxidation is deeper with steam exposure and the external oxide scale becomes thicker. This is most likely due to the accelerated kinetics from grain boundary diffusion of OH- or $H_2O$ [17,41]. There are three possible oxidation mechanisms for the HEA+Cu alloy to form the triplex scale: (1) multiple distinctive oxidation–reduction steps [25], (2) volume expansion from internal spinel formation [25] or (3) activity gradient of oxygen [47,48] driving Cu outward. Unfortunately, the differences for the humid air oxidation do not allow a conclusion to be drawn between the three mechanisms, as accelerated diffusion would be relevant to all. Nevertheless, a comparison with the effect of water vapor on the defect chemistry and growth rate of wüstite [36], along with the observation that the outward diffusion of Cu atoms in $Cu_2O$ determines the growth rate of pure copper [49], leads to the conclusion that, also here, the diffusion of the cations within the oxide is accelerated by the presence

of water vapor. A schematic of the proposed oxidation evolution of the HEA+Cu under humid air at 800 °C is shown in Figure 16.

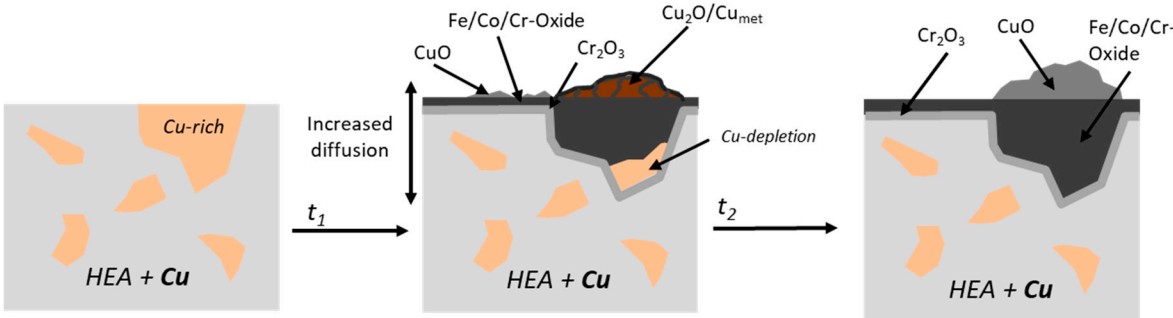

**Figure 16.** Proposed oxidation mechanism of HEA+Cu under humid air exposure at 800 °C as a function of time.

*4.3. HEA+Al*

The two-phase (BCC and FCC) HEA+Al had significantly better oxidation resistance compared to the HEA+Mn and HEA+Cu alloys, especially in the humid atmosphere. As previously noted, the external alumina scale has a needle-like morphology and becomes thicker and rougher, with some AlN formation and Al depletion in the subscale zone, after 300 h and 500 h at 800 °C in air [25]. After water vapor exposure, the formed external $Al_2O_3$ is extremely thin, and a much thinner Al depletion zone becomes evident, showing a very stable, protective oxide layer up to 500 h at 800 °C. Figure 17 summarizes the oxidation behavior of HEA+Al schematically under a humid atmosphere.

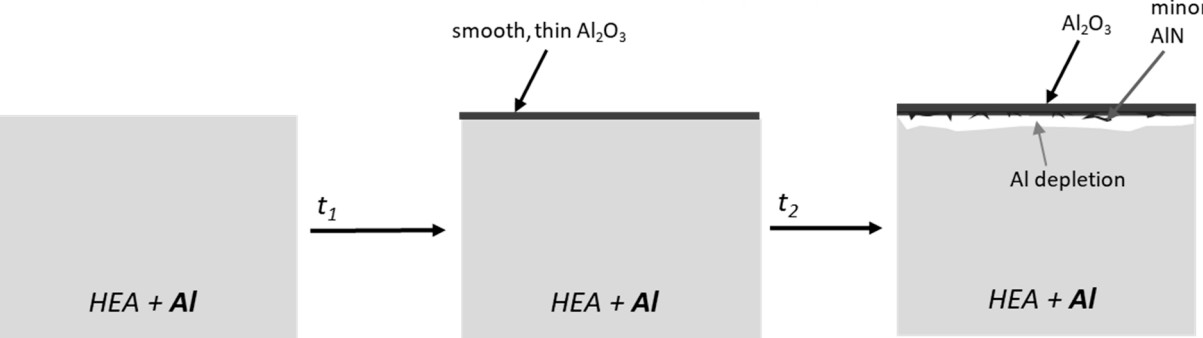

**Figure 17.** Proposed oxidation mechanism of HEA+Al under humid air exposure at 800 °C as a function of time.

The water vapor atmosphere influenced the nitrogen uptake, possibly through a combination of lower nitrogen partial pressure, fewer defects in the alumina scale and/or an accelerated transition from metastable to stable $Al_2O_3$. Additionally, dipole $H_2O$ attaches more easily to the metal surface than $O_2$ and $N_2$ and tends to react more slowly, which then decreases the diffusion of O and N, as the adsorbed $H_2O$ is blocking the adsorption sites on the surface [50–52]. Naumenko et al. observed a similar impact of the atmosphere on TiN formation during the oxidation of a Ni-base alloy [53]. Nowak et al. also detected reduced nitridation of TiN underneath an $Al_2O_3$ layer through the addition of water vapor to the atmosphere [51].

In agreement with the results here, Saunders et al. [20] described faster oxidation rates for most elements, but not Al, when exposed to water vapor. Typically, humidity facilitates the growth of protective hexagonal oxides such as $\alpha$-$Al_2O_3$ over metastable oxides such as $\theta$-$Al_2O_3$ [20] and here, after 500 h under humid air, no $\theta$-$Al_2O_3$ could be observed. Under humid air exposure, the formed $\alpha$-$Al_2O_3$ is less crystallized, indicating additional defects

in the scale, which has also been seen in the literature [33,34]. Often, a visible difference between samples exposed to dry versus humid atmospheres can be seen in the morphology of the oxide scales, as humidity has been observed to facilitate whisker formation [20]. This is the opposite in the current system, where more whiskers and a rougher scale are observed for dry air, suggesting the impact due to the lower nitrogen uptake being higher than the water vapor effect on the growth of the alumina scale. Additionally, the work here suggests a strong influence of water vapor on the stability and prevalence of $\gamma$-$Al_2O_3$, which seems to assist in forming smooth and protective scales.

The thicker Al depletion zone underneath the oxide scale in dry air will eventually become an issue for self-healing of the oxide scale when compared to the humid air exposure. In this case, Cr and spinel would form with increased exposure time and result in the detrimental behavior demonstrated for the Al-free compositions. This HEA+Al alloy shows such extremely protective behavior under water vapor at 800 °C that additional studies at longer exposure durations (to test if $Cr_2O_3$ does eventually form) and at higher temperatures (to examine stability) are of great interest.

*4.4. HEA+Cu/Al*

Under humid air exposure, the HEA+Cu/Al alloy showed excellent oxidation resistance similar to the HEA+Al. While there is a change in the microstructure of the HEA+Cu/Al, as observed previously with the formation of the $\sigma$ phase [25,54], this has minor to no effect on the oxidation behavior in these studies. The amount of Al is sufficient in both the Cu-rich (>5 at.%) and the Cu-lean (>15 at.%) phases to allow protective, external alumina scale formation [25]. One of the most notable differences was the type of alumina that formed on the HEA+Cu/Al: $\theta$ and $\gamma$ were present under dry air and mostly $\gamma$ formed under humid air. In contrast to the HEA+Cu, the alumina scales form on the HEA+Cu/Al before a significant local attack of the Cu-rich phase could occur, forming $(Co,Ni)Fe_2O_4$ spinel. The formation of mixed oxides or spinels (HEA+Cu) was suppressed, and nitridation (HEA+Al) was fully suppressed even after 500 h. The nitridation was seen to be suppressed in the HEA+Cu/Al even in dry air [25], so this protection is compounded by the presence of water vapor occupying adsorption sites on the surface [50,51].

As already shown for the HEA+Al alloy, the morphology of the $Al_2O_3$ scale was affected by the presence of water vapor, shifting from needle-like in dry air to extremely smooth in water vapor. The shift in morphology and shift in crystal structure is most likely the reason for the extremely slow kinetics, with potentially even thinner scales resulting at 500 h for both Al-containing alloys. One explanation could be alumina phase changes or slower crystallization under humid air exposure. Further investigation of the early, transient oxidation of the HEA+Cu/Al alloy would be interesting to see if Cu might act similar to Pt, which is long known to counteract sulfur segregation and void formation [55], which enhances adhesion of alumina scales and lowers the tendency of scale spallation. An increase in scale adhesion is especially important in a water vapor environment, where alumina scales tend to spall more easily [56]. Other oxidation studies of Al and Cu containing HEAs in dry air showed spallation at 1000 °C, which was attributed to a mismatch of the coefficients of thermal expansion (CTE) between the oxide and alloy [57,58]. While spallation was not observed at the 800 °C of this study, further higher-temperature exposures would potentially see this issue, indicating these alloys are potentially better for service at 800 °C and below for their oxidation resistance [59].

As a result of the thinner and smoother scale, the Al consumption, and thus depletion zone, also appears to be slightly less for the HEA+Cu/Al under water vapor, but this must be confirmed through higher-resolution investigations, such as transmission electron microscopy. It is also possible the Al depletion is confounded by the change in microstructure during exposure. The proposed HEA+Cu/Al humid-air oxidation mechanism is schematically shown in Figure 18.

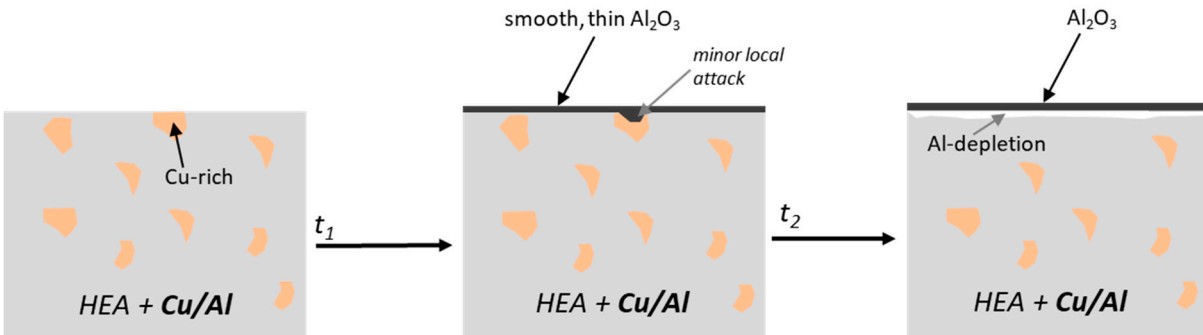

**Figure 18.** Proposed oxidation mechanism of HEA+Cu/Al under humid air exposure at 800 °C as a function of time.

Interestingly, the majority of the detrimental effects of water vapor that were previously listed could not be observed for these HEA alloy systems, except the accelerated growth kinetics of the Cu- and Mn-rich scales via grain boundary diffusion of OH- or $H_2O$ [41], leading to thicker scales (HEA+Mn, HEA+Cu) and higher Kirkendall porosity (HEA+Mn). The exact mechanisms of the alumina-forming HEAs as compared to classical systems under humid air oxidation remain to be fully resolved; however, the scale morphologies clearly change (HEA+Al, HEA+Cu/Al) with water vapor exposure. David Young stated "the situation is complicated by interactions between water vapor and transient alumina phases" [17] and this seems to be even more the case when the substrate is a transition metal high-entropy alloy.

## 5. Conclusions

The humid (10 vol.% $H_2O$) air oxidation behaviors of FeCrMnNiCo (HEA+Mn), FeCrNiCoCu (HEA+Cu), FeCrNiCoAl (HEA+Al) and FeCrNiCoCuAl (HEA+Cu/Al) were investigated at 800 °C for 100, 300 and 500 h.

- All alloys showed variations in their oxidation behavior under the humid atmosphere exposure with the HEA+Al and HEA+Cu/Al demonstrating outstanding oxidation resistance.
- The HEA+Mn formed a fast-growing Mn-containing triplex oxide scale with Kirkendall void formation in the substrate, both of which became worse with longer water vapor exposure.
- The HEA+Cu showed a complex multi-layer oxide scale, which transitioned with longer exposure to CuO/spinel/$Cr_2O_3$. Water vapor accelerated the oxide scale formation, forming a thicker scale, which transitioned sooner, and with deeper internal oxidation of the Cu-rich phase.
- The HEA+Al formed a protective $Al_2O_3$ layer, especially under water vapor exposure, maintaining a thin external scale with a shallow Al depletion zone and less AlN formation than in dry air.
- Lastly, the HEA+Cu/Al also formed a protective $Al_2O_3$ layer with long-term stability and a very minimal local attack of the Cu-rich phase and no AlN formation.
- Both the HEA+Al and HEA+Cu/Al formed smoother oxide scales under humid air exposure, in contrast to needle-like morphologies under synthetic air at 800 °C.

Both HEA alloys with Al showed highly promising behavior when it comes to oxidation behavior in humid atmospheres. This is most relevant for further HEA alloy development and the fundamental understanding of differences in aluminum oxide and nitride formation in water vapor and their dependence on the substrate elements.

**Author Contributions:** Conceptualization, E.M.H.W., C.S. and M.C.G.; Methodology, M.C.G.; Formal analysis, E.M.H.W., C.S., M.B. and M.C.G.; Investigation, M.-L.B. and C.S.; Resources, E.M.H.W., M.B. and M.C.G.; Data curation, M.-L.B.; Writing—original draft, E.M.H.W., M.-L.B. and M.B.; Writing—

review & editing, E.M.H.W., C.S., M.B. and M.C.G.; Visualization, E.M.H.W. and M.B.; Supervision, E.M.H.W. and M.C.G.; Funding acquisition, E.M.H.W. and M.C.G. All authors have read and agreed to the published version of the manuscript.

**Funding:** This research received no external funding.

**Data Availability Statement:** Data are contained within the article.

**Acknowledgments:** EPMA measurements were conducted by Gerald Schmidt at DECHEMA Research Institute. Materials for this study were provided by the "Science-based Acceleration of the Full Value Stream for Metal Additive Manufacturing: Expedited AM Powder Development (X-P4AM)" project funded by the U.S.DOE-EERE-AMO through Ames Lab contract DE-AC02-07CH11358.

**Conflicts of Interest:** The authors declare no conflict of interest.

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
