# Peer review of "High-Temperature Oxidation Behavior of FeCoCrNi+(Cu/Al)-Based High-Entropy Alloys in Humid Air"

_crystals, doi:10.3390/cryst14010060_

Round 1
Reviewer 1 Report
Comments and Suggestions for Authors
In this paper, the authors investigated the high temperature oxidation behavior of FeNiCoCrX alloys (X=Cu, Al, or both) in humid air. The subject is interesting and novel. However, the paper itself does not include any kinetic results. It is a major handicap. Furthermore, the phases are not sufficiently characterized using diffraction methods (X-ray diffraction, electron diffraction). Only SEM results are presented. As such, I cannot recommend this paper for publication in its current form. The following comments should be considered:
1.The paper is submitted to Crystals. As such, the phase constitution of the alloys should be the primary focus. Nevertheless, only SEM and light microscopy images are included in the paper. You must include X-ray diffraction patterns of the alloys before and after oxidation. The phase constitution of the alloys and the oxide scale must be characterized.
2.The SEM images have a low resolution. It is obvious that the oxide was relatively thin (Figs. 6, 8). It should be inspected using HR TEM.
3.A major handicap is the lack of kinetic data. The authors performed the oxidation experiment at 3 different time durations (100, 300, 500 h); however, they haven’t investigated a thickness evolution of the oxide scale over time. It should be studied, and kinetic plots should be presented in the paper. Parabolic rate constants should be obtained and compared with previous studies.
4.The oxide scales of high entropy alloys are often prone to spallation. The reason is a large thermal expansion coefficient mismatch between the alloy and oxide scale, see http://dx.doi.org/10.1002/adem.201500179, http://dx.doi.org/10.1016/j.intermet.2016.12.015, https://doi.org/10.1016/j.jallcom.2018.06.036, etc. This issue is not discussed here.
5.The effect of water vapor is not adequately studied. You should compare the oxidation behavior of the alloys in humid air with those in dry air to draw some general conclusions on the effect of water vapor.
Author Response
In this paper, the authors investigated the high temperature oxidation behavior of FeNiCoCrX alloys (X=Cu, Al, or both) in humid air. The subject is interesting and novel. However, the paper itself does not include any kinetic results. It is a major handicap. Furthermore, the phases are not sufficiently characterized using diffraction methods (X-ray diffraction, electron diffraction). Only SEM results are presented. As such, I cannot recommend this paper for publication in its current form. The following comments should be considered:
1.The paper is submitted to Crystals. As such, the phase constitution of the alloys should be the primary focus. Nevertheless, only SEM and light microscopy images are included in the paper. You must include X-ray diffraction patterns of the alloys before and after oxidation. The phase constitution of the alloys and the oxide scale must be characterized.
X-ray diffraction was conducted; however, for these compositions it is difficult and the results were not included as the florescence was too high making the results not meaningful. This has now been added in the text: “XRD measurements were performed in order to identify the thermally grown oxide scales of all HEAs. Unfortunately, the results were unsatisfactory due to the high fluorescence of the bulk materials. Therefore Raman spectroscopy was pursued instead to identify the oxide scale compositions.” Raman spectroscopy results have now been included for phase composition of the oxide scales in Figures 5, 6, 9 and 12.
2.The SEM images have a low resolution. It is obvious that the oxide was relatively thin (Figs. 6, 8). It should be inspected using HR TEM.
Of course HR-TEM would be highly beneficial, however it was outside of the scope of the current study as the paper was already covering a significant amount of results. It would be interesting to explore in a future paper and is a direction we are considering pending funding. Higher resolution Raman spectroscopy results have now been included for phase composition of the oxide scales in Figures 5, 6, 9 and 12.
3.A major handicap is the lack of kinetic data. The authors performed the oxidation experiment at 3 different time durations (100, 300, 500 h); however, they haven’t investigated a thickness evolution of the oxide scale over time. It should be studied, and kinetic plots should be presented in the paper. Parabolic rate constants should be obtained and compared with previous studies.
Thickness evolution of the oxide scales over time have now been measured and included in kinetic plots in Figure 14. Since the data only includes 3 timesteps, parabolic rate constants were not calculated.
4.The oxide scales of high entropy alloys are often prone to spallation. The reason is a large thermal expansion coefficient mismatch between the alloy and oxide scale, see http://dx.doi.org/10.1002/adem.201500179, http://dx.doi.org/10.1016/j.intermet.2016.12.015, https://doi.org/10.1016/j.jallcom.2018.06.036, etc. This issue is not discussed here.
Spallation was referred to in the discussion of the HEA+Cu/Al: “Further investigation of the early, transient oxidation of the HEA+Cu/Al alloy would be interesting to see if Cu might act similar to Pt, which is long known to counteract sulfur segregation and void formation [48], which enhances adhesion of alumina scales and lowers the tendency of scale spallation. An increase in scale adhesion is especially important in water vapor environment, where alumina scales tend to spall more easily [49].” This has been expanded with the following two sentences and additional references: “Other oxidation studies of Al and Cu containing HEAs in dry air showed spallation at 1000°C which was attributed to a mismatch of the coefficients of thermal expansion (CTE) between the oxide and alloy [50,51]. While spallation was not observed at the 800°C of this study, further higher temperature exposures would potentially see this issue, indicating these alloys are potentially better for service at 800°C and below for their oxidation resistance [52].”
5.The effect of water vapor is not adequately studied. You should compare the oxidation behavior of the alloys in humid air with those in dry air to draw some general conclusions on the effect of water vapor.
This paper was a follow up work to oxidation studies in dry air (reference [25] M.-L. Bürckner, L. Mengis, E.M.H. White, M.C. Galetz, Influence of copper and aluminum substitution on high‐temperature oxidation of the FeCoCrNiMn “Cantor” alloy, Materials and Corrosion 74 (2023) 79–90. https://doi.org/10.1002/maco.202213382). Those previous results are cited in this paper in Table 1, Figures 1, 3, and 5-8 (original draft Figure numbers). Additionally EPMA micrographs comparing dry versus humid air surface/oxide scale cross-sections are included in Figures 2, 4, 6 and 8 (original draft Figure numbers). Throughout the paper these results are compared, starting with the abstract: “The Cu- and Al-containing alloys exhibited improved oxidation resistance over the Mn composition. For the Cu-containing alloy, local attack of the Cu-rich phase was observed which formed a Fe/Ni/Co/Cr spinel that was surrounded by Cr2O3. This oxide was thicker for the humid air atmosphere when compared to dry air and the transition of the Cu oxide to spinel was accelerated. The Al-containing HEA formed a thin Al2O3 scale with humidity suppressing AlN formation and forming a smoother oxide layer. The Al+Cu composition had the highest overall oxidation resistance (minimal local attack, no nitridation) and also showed a smooth oxide scale topography under humid air oxidation as opposed to a plate-like, rougher scale under dry air.”
Reviewer 2 Report
Comments and Suggestions for Authors
The paper titled “High temperature oxidation behavior of FeCoCrNi-based high entropy alloys in humid air” reports the investigation results for the HEA+Mn, HEA+Cu, HEA+Al, and HEA+Cu/Al oxide compositions subjected to the treatment at 800°C in humid air for up to 500 h. It should be noted that in the previous paper [25], the same samples were studied under the dry air conditions. Therefore, the effect of humidity is the only new aspect studied within the current research. To compare these different conditions, many figures from [25] are repeated here. On the other hand, the presented results are interesting and important from practical point of view. Thus, the paper can be recommended for publication after revision. The comments are listed below.
1. Full composition of the synthetic air should be stated.
2. Characterization of the samples should be described in more detail.
3. As is written in the text, the as-solidified chemical compositions and microstructures are listed in Table 1 [24,25]. In the caption of Table 1, work [25] only is referred.
4. Reactions in Table 2 should be numbered, and then Eq. 1 can be eliminated.
5. Local attack is well seen for the HEA+Cu/Al already after 100 h of exposure (Figure 7). This disagrees with the statement that the Al+Cu composition had the highest overall oxidation resistance (no local attack nor nitridation).
Author Response
The paper titled “High temperature oxidation behavior of FeCoCrNi-based high entropy alloys in humid air” reports the investigation results for the HEA+Mn, HEA+Cu, HEA+Al, and HEA+Cu/Al oxide compositions subjected to the treatment at 800°C in humid air for up to 500 h. It should be noted that in the previous paper [25], the same samples were studied under the dry air conditions. Therefore, the effect of humidity is the only new aspect studied within the current research. To compare these different conditions, many figures from [25] are repeated here. On the other hand, the presented results are interesting and important from practical point of view. Thus, the paper can be recommended for publication after revision. The comments are listed below.
- Full composition of the synthetic air should be stated.
This has been added to the experimental section.
- Characterization of the samples should be described in more detail.
Raman spectroscopy results have been added for the scale characterization, as well as kinetic plots of scale growth. These are described in the Experimental Methods.
- As is written in the text, the as-solidified chemical compositions and microstructures are listed in Table 1 [24,25]. In the caption of Table 1, work [25] only is referred.
Reference 24 has now also been included in the caption of Table 1.
- Reactions in Table 2 should be numbered, and then Eq. 1 can be eliminated.
The text was slightly modified so that equation 1 could be eliminated, in favor of referencing Table 2.
- Local attack is well seen for the HEA+Cu/Al already after 100 h of exposure (Figure 7). This disagrees with the statement that the Al+Cu composition had the highest overall oxidation resistance (no local attack nor nitridation).
This has been modified in the abstract to “The Al+Cu composition had the highest overall oxidation resistance (minimal local attack, no nitridation)…” which is now consistent with the rest of the paper.
Reviewer 3 Report
Comments and Suggestions for Authors
Please address the following comments in the revised manuscript:
1-What are the specific compositions of the FeCoCrNi-based high entropy alloys (HEAs) mentioned in the study, including those with Mn, Cu, Al, or Al+Cu additions?
2-How does the presence of water vapor (humid air) affect the oxidation behavior of these HEAs compared to dry air at 800°C?
3-What are the observed differences in oxidation resistance among the different alloy compositions, especially regarding the formation of oxide scales and any local attack?
4-Can you explain the mechanisms behind the formation of various oxide scales, such as CuO, spinel, Cr2O3, and Al2O3, in different alloy compositions under humid air exposure?
5-How does the presence of water vapor influence the growth and stability of oxide scales in these HEAs, and what is the role of water vapor in these processes?
6-What are the implications of these findings for potential high-temperature applications of these HEAs in humid atmospheres, and what further research directions are suggested based on the results?
7-Extension of the reference part is recommended. The following papers can be helpful:
[a] Jom 67, 2015, 2326-2339.
[b] Journal of Alloys and Compounds 954, 2023, 170091
[c] Advanced Engineering Materials 23, no. 5, 2021, 2001047
Author Response
1-What are the specific compositions of the FeCoCrNi-based high entropy alloys (HEAs) mentioned in the study, including those with Mn, Cu, Al, or Al+Cu additions?
The as-solidified compositions are included in Table 1.
2-How does the presence of water vapor (humid air) affect the oxidation behavior of these HEAs compared to dry air at 800°C?
The abstract states, “The Cu- and Al-containing alloys exhibited improved oxidation resistance over the Mn composition. For the Cu-containing alloy, local attack of the Cu-rich phase was observed which formed a Fe/Ni/Co/Cr spinel that was surrounded by Cr2O3. This oxide was thicker for the humid air atmosphere when compared to dry air and the transition of the Cu oxide to spinel was accelerated. The Al-containing HEA formed a thin Al2O3 scale with humidity suppressing AlN formation and forming a smoother oxide layer. The Al+Cu composition had the highest overall oxidation resistance (minimal local attack, no nitridation) and also showed a smooth oxide scale topography under humid air oxidation as opposed to a plate-like, rougher scale under dry air.” The oxidation in water vapor versus that in dry air at 800C is compared throughout the results and discussion sections as it is the main focus of the paper. The exact influence of water vapor varies by composition.
3-What are the observed differences in oxidation resistance among the different alloy compositions, especially regarding the formation of oxide scales and any local attack?
The Conclusion states, “The alloys showed variations in their oxidation behavior under the humid atmosphere exposure with the HEA+Al and HEA+Cu/Al demonstrating outstanding oxidation resistance. The HEA+Mn formed a fast growing Mn-containing triplex oxide scale with Kirkendall void formation in the substrate, both of which became worse with longer water vapor exposure. The HEA+Cu also showed a complex multi-layer oxide scale which transitioned with longer exposure to CuO/spinel/Cr2O3. Water vapor accelerated the oxide scale formation, forming a thicker scale which transitioned sooner, and with deeper internal oxidation of the Cu-rich phase. The HEA+Al formed a protective Al2O3 layer, especially under water vapor exposure, maintaining a thin external scale with a shallow Al depletion zone and minimal AlN formation. Lastly the HEA+Cu/Al also formed a protective Al2O3 layer with long term stability and very minimal local attack of the Cu-rich phase. Both the HEA+Al and HEA+Cu/Al formed smoother oxide scales under humid air exposure, in contrast to needle-like morphologies under synthetic air at 800°C.”
4-Can you explain the mechanisms behind the formation of various oxide scales, such as CuO, spinel, Cr2O3, and Al2O3, in different alloy compositions under humid air exposure?
The Discussion states, “The exact mechanisms of the alumina-forming HEAs as compared to classical systems under humid air oxidation remain to be fully resolved, however the scale morphologies clearly change (HEA+Al, HEA+Cu/Al) with water vapor exposure. David Young stated “the situation is complicated by interactions between water vapor and transient alumina phases” [17] and this seems to be even more the case when the substrate is a transition metal high entropy alloy.”
The remaining oxide formation can be explained by Table 2, with the list of partial pressures for hydroxide formation, as well as well-known Gibbs free energies of oxide formation (Arrhenius plot). Lastly the Discussion includes, “There are three possible oxidation mechanisms for the HEA+Cu alloy to form the triplex scale: 1) multiple distinctive oxidation-reduction steps [25], 2) volume expansion from internal spinel formation [25], or 3) activity gradient of oxygen [40,41] driving Cu outward. Unfortunately the differences for the humid air oxidation do not allow a conclusion to be drawn between the three mechanisms, as accelerated diffusion would be relevant to all.”
5-How does the presence of water vapor influence the growth and stability of oxide scales in these HEAs, and what is the role of water vapor in these processes?
While differences can be observed amongst the compositions and the atmospheres, the specific role of water vapor cannot be explicitly stated based on these observations as multiple roles are theoretically possible. The paper attempts to summarize this at the beginning of the Discussion section with “water vapor has a complicated effect on high temperature oxidation behavior including: increasing proton defects in the oxide scale [26],[27], accelerating the transformation of θ- to α-Al2O3 [28], forming H2–H2O (facilitating rapid inward transport of O across pores) [29],[30], and increasing the vaporization of some oxides by forming hydroxides [28].” This is one of the reasons water vapor exposures are so interesting and motivated the present work, as the specific mechanisms are still unclear and potentially compounded in complex alloys.
6-What are the implications of these findings for potential high-temperature applications of these HEAs in humid atmospheres, and what further research directions are suggested based on the results?
The Conclusion states, “Both the HEA+Al and HEA+Cu/Al formed smoother oxide scales under humid air exposure, in contrast to needle-like morphologies under synthetic air at 800°C. Both of these alloys are promising alloys to be developed for high temperature applications in humid atmospheres, as well as for further elucidating the underlying oxidation mechanisms.”
The discussion states, “This HEA+Al alloy shows such extremely protective behavior under water vapor at 800°C, that additional studies at longer exposure durations (to test if Cr2O3 does eventually form) and at higher temperatures (to examine stability) are of great interest.”
The discussion section also includes, “As a result of the thinner and smoother scale, the Al consumption, and thus depletion zone, also appears to be slightly less for the HEA+Cu/Al under water vapor, but this must be confirmed by higher resolution investigations, such as transmission electron microscopy.”
These are the more specific implications and future research directions pertinent to this alloy family, but in general water vapor-containing atmospheres are of great research interest for alumina formers as highlighted by the statement “The exact mechanisms of the alumina-forming HEAs as compared to classical systems under humid air oxidation remain to be fully resolved, however the scale morphologies clearly change (HEA+Al, HEA+Cu/Al) with water vapor exposure. David Young stated “the situation is complicated by interactions between water vapor and transient alumina phases” [17] and this seems to be even more the case when the substrate is a transition metal high entropy alloy.” in the discussion.
7-Extension of the reference part is recommended. The following papers can be helpful:
[a] Jom 67, 2015, 2326-2339. This is already included as reference 9.
[b] Journal of Alloys and Compounds 954, 2023, 170091. While this paper is quite interesting and impactful in the HEA literature, it addresses the composition CrMnFeNiCu, which is both Cu- and Mn-containing, without Co, and thus it is difficult to make a direct comparison with the current paper, especially as the Mn alloy was only included as a baseline and Mn is detrimental to the oxidation resistance (as highlighted by the above JOM paper reference 9).
[c] Advanced Engineering Materials 23, no. 5, 2021, 2001047. This is already included as reference 14.
The following references were added, along with those for the Raman results:
[50] H.M. Daoud, A.M. Manzoni, R. Völkl, N. Wanderka, U. Glatzel, Oxidation Behavior of Al 8 Co 17 Cr 17 Cu 8 Fe 17 Ni 33 Al 23 Co 15 Cr 23 Cu 8 Fe 15 Ni 15 and Al 17 Co 17 Cr 17 Cu 17 Fe 17 Ni 17 Compositionally Complex Alloys (High‐Entropy Alloys) at Elevated Temperatures in Air, Adv Eng Mater 17 (2015) 1134–1141. https://doi.org/10.1002/adem.201500179.
[51] J. Dabrowa, G. Cieølak, M. Stygar, K. Mroczka, K. Berent, T. Kulik, M. Danielewski, Influence of Cu content on high temperature oxidation behavior of AlCoCrCuxFeNi high entropy alloys (x-á=-á0; 0.5; 1), Intermetallics 84 (2017) 52–61.
[52] L. Chen, Z. Zhou, Z. Tan, D. He, K. Bobzin, L. Zhao, M. Öte, T. Königstein, High temperature oxidation behavior of Al0.6CrFeCoNi and Al0.6CrFeCoNiSi0.3 high entropy alloys, Journal of Alloys and Compounds 764 (2018) 845–852. https://doi.org/10.1016/j.jallcom.2018.06.036.
Round 2
Reviewer 1 Report
Comments and Suggestions for Authors
The authors answered my previous comments and explained the lack of XRD results by the fluorescence of the substrate. The paper has been improved. It is publishable subject to minor (formal) revision:
1.You should not present results in conclusions (Fig. 14 on page 22). The figure on p. 22 should be removed, as the same results (oxide layer thicknesses) are already presented in Fig. 14 on p. 16. Furthermore, there cannot be two figures with identical numbers (Fig. 14) in the same manuscript.
2.The conclusions (p. 22) should be given point-by-point and numbered consecutively. It will improve the readability of the conclusions.
Author Response
- The figures have been adjusted/deleted accordingly.
- The main conclusions have been changed to a bulleted list.
Reviewer 2 Report
Comments and Suggestions for Authors
The authors have revised the manuscript satisfactorily. The reviewers’ comments are well addressed. The paper can be accepted for publication in its current form.
Author Response
The authors thank the reviewer for their time and efforts.
Reviewer 3 Report
Comments and Suggestions for Authors
The revision is acceptable for publication
Author Response

(The authors gave the same response as above.)
